# Application of Skyline for Analysis of Protein–Protein Interactions In Vivo

**DOI:** 10.3390/molecules26237170

**Published:** 2021-11-26

**Authors:** Arman Kulyyassov

**Affiliations:** Republican State Enterprise “National Center for Biotechnology” under the Science Committee of Ministry of Education and Science of the Republic of Kazakhstan, 13/5, Kurgalzhynskoye Road, Nur-Sultan 010000, Kazakhstan; kulyyasov@biocenter.kz; Tel.: +7-7172-707534

**Keywords:** biotin acceptor peptide (BAP), biotin ligase BirA, liquid chromatography tandem mass spectrometry (LC-MS/MS), multiple reaction monitoring (MRM), protein–protein interactions (PPIs), proximity utilizing biotinylation (PUB), proteomics

## Abstract

Quantitative and qualitative analyses of cell protein composition using liquid chromatography/tandem mass spectrometry are now standard techniques in biological and clinical research. However, the quantitative analysis of protein–protein interactions (PPIs) in cells is also important since these interactions are the bases of many processes, such as the cell cycle and signaling pathways. This paper describes the application of Skyline software for the identification and quantification of the biotinylated form of the biotin acceptor peptide (BAP) tag, which is a marker of in vivo PPIs. The tag was used in the Proximity Utilizing Biotinylation (PUB) method, which is based on the co-expression of BAP-X and BirA-Y in mammalian cells, where X or Y are interacting proteins of interest. A high level of biotinylation was detected in the model experiments where X and Y were pluripotency transcription factors Sox2 and Oct4, or heterochromatin protein HP1γ. MRM data processed by Skyline were normalized and recalculated. Ratios of biotinylation levels in experiment versus controls were 86 ± 6 (3 h biotinylation time) and 71 ± 5 (9 h biotinylation time) for BAP-Sox2 + BirA-Oct4 and 32 ± 3 (4 h biotinylation time) for BAP-HP1γ + BirA-HP1γ experiments. Skyline can also be applied for the analysis and identification of PPIs from shotgun proteomics data downloaded from publicly available datasets and repositories.

## 1. Introduction

Wide practical application of liquid chromatography in combination with mass spectrometry has been observed recently in proteomics [1,2] and metabolomics [3,4] as a routine method for the qualitative and quantitative analysis of biological samples. For example, when optimizing expression, performing quality control, or studying pharmacokinetics of recombinant proteins, it is crucial that the best conditions for production or analysis of the drug products are found [5,6]. Another important task is to obtain information about changes in the expression of marker proteins under different physiological conditions of the cell [7]. Examples of this include: differences in protein composition in a healthy/cancer cell or differences under the influence of external factors such as temperature, chemical agents, or radiation provide valuable information about metabolic and signaling pathways, mechanisms of stress response. In all these cases, the results are obtained as chromatograms in the multiple reaction monitoring (MRM) method, where many peptides derived from target proteins can be identified by retention time and mass spectra of fragment ions (or MS/MS spectra), and the relative amount of each peptide between samples can be determined by comparison of the peak areas [8,9,10].

However, information on protein composition is not sufficient to fully understand the mechanism of cell function. The quantification of protein–protein interactions (PPIs) in vivo can be a useful extension in research since more than 80% of proteins do not function separately, but rather interact and participate in the formation of stable or transient complexes [11]. These protein–protein interactions play an important role in almost all vital processes in cells, such as DNA replication, gene transcription and translation, signal transduction, cell-cycle control and proliferation, and cell–cell communication [12].

Methods based on a combination of affinity purification (AP) or tandem affinity purification (TAP) and mass spectrometry (MS/MS) have now become standard for the identification of protein partners [13,14,15,16]. However, these methods have the serious drawback of a large number of false-positive identifications [17]. In addition, the cell lysis procedure can lead to the destruction of weak protein–protein interactions, which can also lead to false-negative results. For example, the list of Oct4-interacting proteins identified by co-immunoprecipitation (Co-IP) did not include one of the most studied Oct4 partners, namely Sox2 [18].

Enzymatically catalyzed proximity labeling is an alternative to immunoprecipitation and biochemical fractionation for the proteomic analysis of macromolecular complexes and protein interaction networks [19]. In this method, ligation enzymes are expressed in cells as conjugates with proteins of interest. For example, proximity-dependent biotinylation methods are based on the use of mutant biotin ligases, BioID [20] or TurboID [21]. These BirA mutants prematurely release the highly reactive yet labile biotinoyl-AMP inside of a living cell, which readily reacts with lysine’s primary amino groups of proximal proteins.

On the other hand, the proximity utilizing biotinylation (PUB) method is based on the use of humanized wild-type biotin ligase BirA fusions. The wild-type BirA uses biotin and ATP to generate biotinoyl-AMP [22,23]. Wild-type BirA holds on to the reactive biotin molecule until it is covalently attached to a very specific substrate called biotin acceptor peptide (BAP) [24]. Thus, biotinylation is a result of the direct contact of BirA and BAP parts of recombinant proteins which occurs in cases of protein–protein interaction or random collision in vivo.

Both methods are based on similar principles; this is the in vivo creation of a permanent covalent mark on one of the proteins of interest or partners interacting with them, which allows us to bypass the limitations imposed by the extraction and purification stages. Ultimately, results will be obtained with much fewer false-positive and false-negative protein identifications compared to traditional methods such as IP-MS/MS or TAP. The result is the facilitation of the bioinformatic part of data analysis.

The aim of this work was to use the Skyline program to process the results of experiments on the quantitative analysis of PPI using the proximity utilizing biotinylation (PUB) method.

## 2. Results and Discussion

### 2.1. Overview of the Method and Experimental Workflow of the PUB Protocol

The principle of the PUB method is based on using enzyme/substrate pair reactions [25,26,27,28], where two proteins to be tested for their interaction in vivo are co-expressed in mammalian cells, one as fused to the BAP, and the other fused to an enzyme BirA, which is an *Escherichia coli* protein biotin ligase [29]. When the two proteins are in proximity to each other, for example, when an interaction of X and Y occurs in vivo, a more efficient biotinylation of the BAP is to be expected (Figure 1A). The biotinylation status of the BAP fusion protein can be further monitored by Western blot, mass spectrometry, or confocal microscopy (Figure 1B,C). HEK293T, HeLa, or MRC-5 fetal lung fibroblast cell lines can be used for the transient or stable expression of recombinant proteins in the PUB method. Usually, one control experiment is performed in a parallel dish or a 6-well plate, using cells in which non-interacting proteins or other pair proteins are expressed for comparison (BirA-X and BAP-Z). Depending on the proteins of interest chosen for the experiment, biotin is added to the medium from 5 min to 9 h before harvesting the cells. The sequence of the BAP peptide was modified and compared to commonly used peptides, such as Avitag [30,31], in order to reduce the level of background biotinylation [28]. Additionally, 7His-tag was added upstream of the sequence to provide the option to purify both labeled and nonlabeled BAP fusion proteins from cell lysates.

The experimental workflow for the LC-MS/MS analysis of samples includes additional steps, such as the purification of Ni agarose beads, propionylation, and on-gel (or on-bead) tryptic digest (Figure 1C). Propionylation was used to protect the nonbiotinylated BAP peptide from tryptic cleavage on the target lysine. This modification resulted in the production of modified and nonmodified peptides of comparable sizes, facilitating the interpretation of results. After analysis on Skyline (Figure 1D), data were exported as CSV files and processed using Microsoft Excel for the calculation of biotinylation levels (Appendix A). First, the total amount of BAP was calculated by the addition of the total area of propionylated BAP to the total area of recalculated biotinylated BAP. For the recalculation of the biotinylated BAP, the relative ionization coefficient k = 11.9 was used (Figure 1E), which was estimated earlier in SILAC experiments [28]. The ionization efficiency depends on the chemical structure of a molecule and would thus be different for propionyl and biotin residues. Therefore, a direct comparison between total ion chromatograms (TIC) of the biotinylated and propionylated BAP in LC-MS/MS data is not possible. After this step, the areas were normalized, and normalization coefficients of the total amounts of BAP were calculated for each sample. These normalization coefficients were then used to recalculate the biotinylation levels and for the estimation of means and standard deviations.

The total amount of BAP was calculated using the formula A_BAP_ = k × A_bBAP_ + A_pBAP_, where A_bBAP_ corresponds to the peak area of total ion chromatograms (TIC) of biotinylated, and A_pBAP_ to propionylated BAP, and k is the relative ionization coefficient between the biotinylated and propionylated BAP peptides (Figure 1D). The chromatographic elution peaks of the fragments for the four most intensive ions, *y*_7_, *y*_6_, *y*_5_, and *y*_4_, in extracted ion chromatograms (EIC) were integrated and summed to give the peak area of TIC. In the control (BAP-Y + BirA-Z) and in the experiment (BAP-Y + BirA-X), as well as in the replicates, the expression levels of recombinant BAP-Y proteins may differ. Variations in the total amount of BAP-Y can also appear during a sample preparation. Thus, direct comparison of the biotinylation levels A_bBAP_ between samples is not correct. Therefore, the total amount of BAP-Y, including its biotinylated and propionylated forms in all samples, was normalized and the normalization coefficients were determined (Appendix A).

The Skyline is an application for targeted proteomics and quantitative data analysis in the frame of the Windows operation system [32]. Its interface facilitates the improvement of mass spectrometer methods and the analysis of data from targeted MRM experiments. Skyline imports the native output files from instruments manufactured from different vendors smoothly, connecting mass spectrometer output back to the experimental design document. A rich choice of graphics displays provide powerful tools for inspecting and monitoring data integrity as data are acquired, helping instrument operators to identify problems early. It is open-source and freely available for commercial and academic use [9,33]. In addition, its output data format (csv.files) allows the performance of post-processing analysis in Microsoft Excel to recalculate biotinylation levels.

This software was successfully used to identify and quantify the target BAP peptides from all MRM data (Appendix A). Since the amino acid sequence of the BAP1070 peptide is artificially generated and is absent in the NCBI and Swissprot databases, the sequence of this peptide was added to a client-made database, BAP1070, using the Database manager (Figure 2A). This allowed the DAT file to be generated and the spectral libraries to be created in Skyline.

Prior to importing raw data into Skyline, a spectral library containing the product ion spectra of the BAP target peptides was constructed using the DAT file. The spectral library consisted of MS/MS spectra of biotinylated and propionylated forms of BAP1070 peptide. A spectral library allowed for the direct comparison of BAP target peptide product ion spectra from the MRM analyses to the corresponding product ion “library match”. Product ion transitions used to confirm the identity of each target peptide in the MRM analyses were automatically picked based on the four most abundant y-type product ion intensities observed in the “library match” spectrum.

#### 2.1.1. Creation of MRM Method and LC-MS/MS Analysis of the Samples

The vendors’ default method (Bruker Company) was used for the creation of the MRM method, as described earlier [34]. Precursor ions: *m*/*z* 563.2 (ILEAQK(Prop)IVR) propionylated form of BAP, and *m*/*z* 648.8 (ILEAQK(Biot)IVR) biotinylated form of BAP.

#### 2.1.2. Creation of BAP1070 Database on Mascot Search Server Using Database Manager

Before analysis on Skyline, the raw LC-MS/MS data were processed to a special format—DAT file. First, the raw data were analyzed on Bruker DataAnalysis software to generate an MGF file. Then, the BAP1070 database was created containing a sequence of this peptide in the Database manager (Figure 2A), which is a browser-based utility for updating and configuring local copies of sequence databases. Analysis of an MGF file against the BAP1070 database in the Mascot search engine, including propionylation and biotinylation modifications, yielded a report where results could also be exported as a DAT file (Figure 2B).

#### 2.1.3. MRM Analysis and Post-Processing of Data

All MRM data were analyzed in Skyline 19.1.0.193. A spectral library was constructed from the peptide identifications from a DAT file exported from the Mascot result page. The four product ions extracted by Skyline were determined based on the ranking of the top four most intense y-ions from the corresponding library spectrum for each peptide. Dot-product (dotp) scores were calculated based on the correlation of the measured product ion peak intensities with the peak intensities observed in the library spectrum for that same peptide [33]. Raw LC-MS/MS data and processed files were uploaded to the Panorama repository [35].

### 2.2. DNA Dependent Interaction of Sox2 and Oct4

After processing and recalculating Skyline results, quantitative data on biotinylation levels were obtained (Figure 3A,B). The samples from experiments with the co-expression of BAP-Sox2 and BirA-Oct4 in HEK293T cells showed a high level of biotinylation (86 ± 6 and 71 ± 5 for different biotinylation times). This is due to the presence of DNA binding domains, HMG present in Sox2 and POU in Oct4, which recognize *Utf1* or other motifs [36] and result in close contact between target BAP and biotin ligase BirA. Contrary to Sox2, GFP lacks DNA binding domains, and in a control experiment with the coexpression of BAP-GFP and BirA-Oct4, very low biotinylation levels were observed (sample 0–9 on Figure 3A). Recombinant proteins BAP-GFP and BAP-Sox2 from HEK293T cell nuclear lysates were purified on Ni sepharose beads and propionylated before trypsin digest, as described earlier [34].

### 2.3. Protein Oligomerization HP1γ-HP1γ

Heterochromatin protein HP1γ was chosen as another example of protein–protein interactions. The proteins of this family contain a chromo shadow domain (CSD), which allows them to form dimers and oligomers [37,38]. The formation of these oligomeric structures is critical for the organization of heterochromatin in the cell nucleus [39]. In the experiment, BAP-HP1γ and BirA-HP1γ protein pairs were expressed. Another protein, TAP54α, participates in the formation of hexamers with ATPase activity and is a component of histone acetyltransferase complexes [40,41]. Since Tap54α is not a protein that interacts with HP1γ, we chose a model where other protein pairs, BAP-HP1γ and BirA-Tap54α, were expressed in a separate dish as a control. The difference in the biotinylation level of the experiment (BAP-HP1γ + BirA-HP1γ) versus control (BAP-HP1γ + BirA-Tap54α) after processing the raw mass spectrometer data with Skyline was also significant and was 32 ± 3 (Figure 3C,D). The raw data were obtained on an Agilent nanoHPLC-Chip-3D6340 Iontrap instrument and converted to mzML format [42] using ProteoWizard software [43,44].

### 2.4. Analysis of Shotgun Proteomics Data (Mutant Biotin Ligase BioID Application)

The Skyline program was mainly developed for the analysis of MRM results in targeted proteomics [32]. However, this application can also be used to analyze the results of Shotgun proteomics where an instrument is operated in data-dependent acquisition (DDA) mode. For validation, the raw data from the results of the BioID mutant biotin ligase experiments, published recently by Go et al. [45] and publicly available in the Massive repository, were downloaded. Go et al. identified 35,902 interactions with 4424 unique high-confidence proximity interactors for 192 BioID fused bait proteins from different cellular compartments. The MGF file was used to obtain the DAT file, as described earlier in Section 2.1.2, which was used in Skyline to build the spectral library. Since these results were obtained on a different instrument platform (Eksigent NanoLC-Ultra 2D plus HPLC system-Orbitrap Elite) and under different modes of operation, the parameters in the Peptide setting and Transition settings tabs were changed, as described in the experimental part from paper [45]. An example of interacting (or proximal) proteins of mitochondrial Pyruvate Dehydrogenase E1 Subunit Alpha 1 (PDHA1) fused with BioID is shown in Figure 4.

### 2.5. Perspectives of Enzymatically Catalyzed Labeling for Biological Research

Methods based on the use of wild-type biotin ligase BirA and their mutant versions, BioID or TurboID, have a similar principle: the creation of a permanent covalent label on the partner protein in vivo. This facilitates subsequent steps in protocols and especially easy and efficient purification of biotinylated proteins from cell lysates using commercially available reagents and kits.

In addition, these methods can be complementary to each other. For example, while the use of BioID or TurboID allows the identification of proximal or partner proteins in cells, the PUB method can be used to quantitatively compare the identified protein–protein proximity.

The MRM method, where the output results are presented as coupled data of chromatographic parameters and mass spectra for each peptide, is now widely used to study the mechanisms of external influences (temperature, radiation, or chemical reagents) on the expression of various proteins in a cell. By analogy, the PUB method can be used to study the influence of various external factors on protein–protein interactions in a living cell, which can extend a given research area and provide additional information about cell organization and function.

Thus, the use of modern bioinformatics programs such as Skyline in combination with PUB, BioID, and TurboID methods will facilitate the analyses of large amounts of data to solve various problems of cell and molecular biology.

## 3. Materials and Methods

Cell culture, transient transfection, and sample preparation steps were described earlier [28,34].

The peptide mixtures were analyzed using two LC-MS/MS systems:Nanoflow HPLC system (Thermo Dionex Ultimate 3000, ThermoScientific) with Acclaim PepMap100 C18 pre-column, 5 mm × 300 μm; 5 μm particles (Thermo Scientific, #160454) and Acclaim Pep-Map RSLC column 15 cm × 75μm, 2 μm particles (Thermo Scientific, #164534) coupled via CaptiveSpray to the QTOF Impact II mass spectrometer (Bruker). Raw LC-MS/MS data were interpreted with the Bruker Compass DataAnalysis (version 4.3) software. The separation gradient was 48 min from 2% to 50% acetonitrile. Flow rates—300 nL/min.Nano-HPLC (Agilent Technologies 1200) was coupled to an ion-trap mass spectrometer (Bruker 6300 series) equipped with a nanoelectrospray source via protein HPLC Chip (Agilent Technologies, G4240-62001) with 40 nL trap 75 um × 43 mm 5 um 300SB-C18-ZX and analytical column packed with ZORBAX 300SB-C18, 5 µm particle size. The separation gradient was 7 min from 5% to 90% acetonitrile. Flow rates—300 nL/min.

The LC-MS/MS instruments were set to monitor transitions of biotinylated (*m*/*z* 648.8, collision energy 33.0 eV) and propionylated (*m*/*z* 563.2, collision energy 27.0 eV) forms of BAP peptide in samples.

### 3.1. Data Preparation and Creation of MGF File Using DataAnalysis

For preliminary analysis of data and generation of peak lists, DataAnalysis (DA 4.1) software was used. Retention-time information was changed to seconds.

The MGF file was generated from raw data by clicking the following tabs on the menu: Find/Compounds MS(n)→Deconvolute/Mass spectra → File/Export/Compounds. Subsequent database searches were performed using Mascot search engine. Then, the results were imported as the DAT file which were used to build a spectral library in Skyline.

### 3.2. Creation of BAP1070 Fasta Database on Mascot Search Server

In MS Notepad text editor, the aminoacid sequence of BAP1070 was pasted with the description line as follows:

>BAP1070

GHHHHHHHGLTRILEAQKIVRGG

This file was saved as BAP1070_fasta.txt

On mascot server http://mascot-server/mascot/index.html (Configuration last updated Thu Apr 15 10:36:27 2021), the BAP1070 database was created using the following steps: Home subpage → MascotUtilities → ConfigurationEditor → Database Manager → Fasta → Create new. Configuration details for BAP1070: Database name—BAP1070, Database type—aminoacid, Accession parse rule— > [^] *\(.*\), Description parse rule > [^] *\(.*\), Taxonomy source—none, Sequence report source—FASTA file, Full-text report source—None, Number of threads—automatic, Use memory mapping?—Yes, Lock to memory?—No.

Analysis of data, including Building a Spectral Library in Skyline, Configuring Transition Settings, Populating the Skyline Peptide Tree, Importing Raw Data into Skyline and Subsequent Filtering, and data processing and calculation of biotinylation levels are described in Appendix A.

## 4. Conclusions

In this study, the Skyline program was used for the first time to analyze results obtained by using a proximity utilizing biotinylation method based on expression in mammalian target cells BAP-X and wild-type BirA-Y protein conjugates (first example: X-Sox2, Y-Oct4, versus control X-GFP, Y-Oct4 and second: example X, Y-HP1γ versus control X-HP1γ, Y- Tap54α). Peak areas of biotinylated BAP were used for the estimation of PPI, while peak areas of propionylated BAP on MRM chromatograms were used for the recalculation and normalization of data between different samples. This program allowed for fast processing of raw data, the calculation of peak areas, and provided the output file in CSV format, which is convenient for subsequent analysis on Microsoft Excel.

Skyline was also used to analyze data on protein–protein interactions and proximities obtained by using mutant biotin ligase BioID [45]. These raw data were downloaded from the MassIVE Repository database and were sourced from another LC-MS/MS instrument platform, demonstrating that the Skyline program is not “instrument or vendor-oriented”.

Overall, the Skyline program offers an advantage in that it provides a good graphic representation of data and reduces analysis time. This protocol could be applicable, not only to BAP, but also to other synthetic peptides which are absent in NCBI or SwissProt databases.

## Figures and Tables

**Figure 1 molecules-26-07170-f001:**
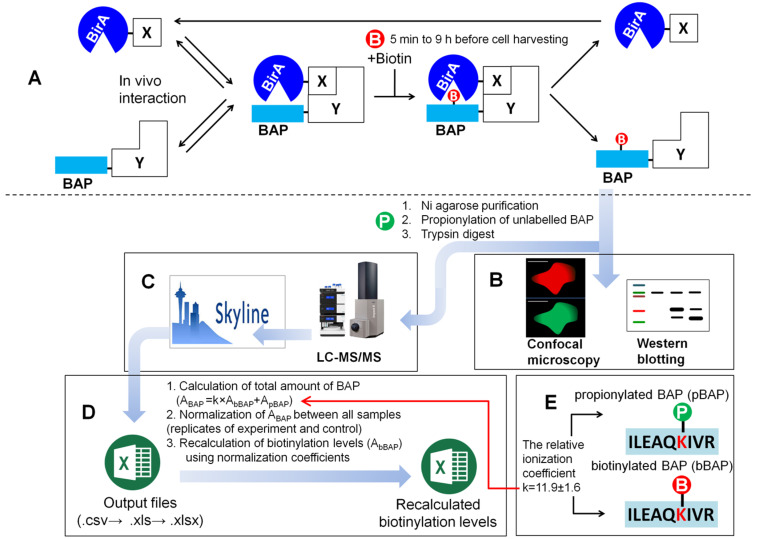
Principle of PUB method in living cell and workflow for quantification of PPI. In vivo interaction (**A**) of proteins X and Y results in site-specific biotinylation of the biotin acceptor peptide (BAP) by wild-type humanized biotin ligase (BirA). Biotinylated protein can be detected by WB, for example, Streptavidin-HRP, IF confocal microscopy (**B**) or LC-MS/MS (**C**). X or Y–HP1(α,β,γ), Tap54(α,β), Sox2, Oct4, or other proteins. Biotinylation levels of BAP peptides obtained after processing the results of sample analyses using Skyline (**C**) are recalculated in Microsoft Excel (**D**). P—propionylated form of BAP1070: GHHHHHHHGLTR**I****LEAQK(Prop)IVR**GG, B—biotinylated form of BAP1070: GHHHHHHHGLTR**ILEAQK(Biot)IVR**GG, the sequence corresponding to the peptide on the chromatogram after trypsin digest is marked in bold. The relative ionization coefficient of tryptic peptides derived from propionylated and biotinylated BAP (**E**).

**Figure 2 molecules-26-07170-f002:**
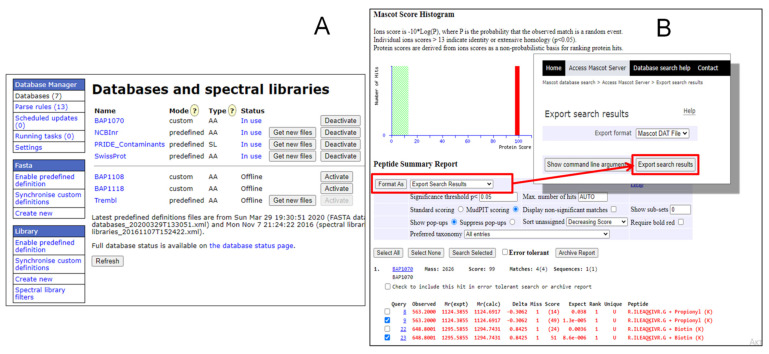
Creation of BAP1070 file. (**A**) Screenshots of the database manager page with BAP1070 file, (**B**) Mascot search results and exporting the DAT file.

**Figure 3 molecules-26-07170-f003:**
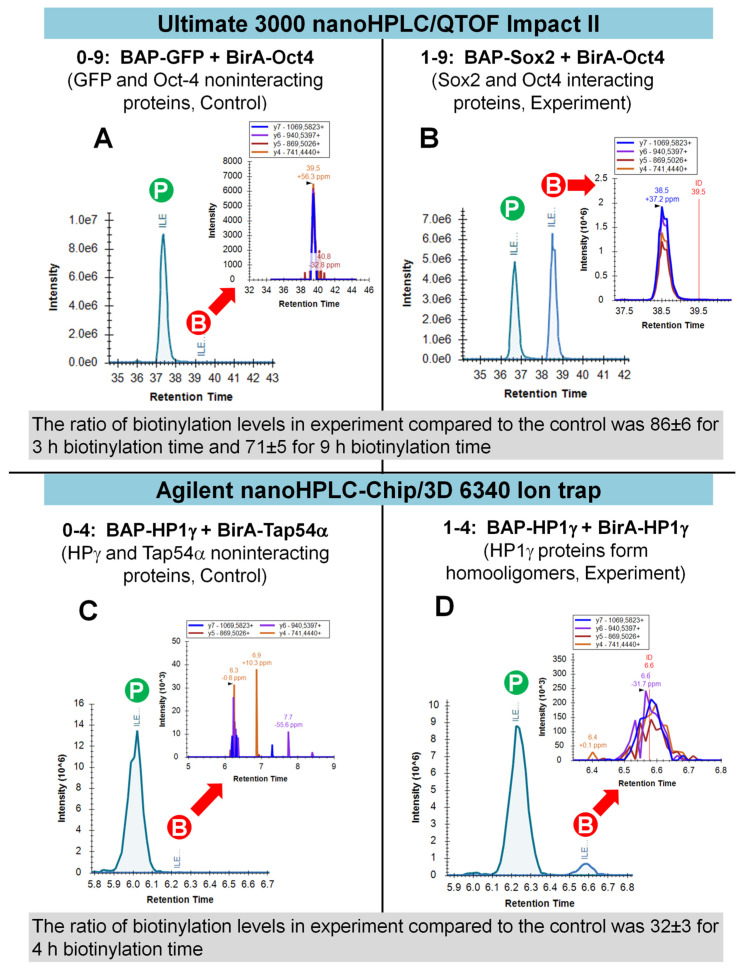
Skyline graphical representation of chromatograms from PUB experiments. Total ion chromatograms of propionylated (P in green circle) and biotinylated (B in red circle) BAP peptides for two examples of experimental PPIs (**B**,**D**) obtained using different instrument platforms (**A**–**B**, **C**–**D**). Four most intense fragment ions, *y*_7_, *y*_6_, *y*_5_ and *y*_4_, were chosen for area calculation of biotinylated BAP peptide in extracted ion chromatograms. Left side are controls BAP-GFP + BirA-Oct4 (**A**) and BAP-HP1γ + BirA-Tap54α (**C**) and right side of the figure represents experiments with interacting proteins—BAP-Sox2 + BirA-Oct4 (**B**) and BAP-HP1γ + BirA-HP1γ (**D**). The average ratios of biotinylation levels were obtained from three experiments after recalculation and normalization.

**Figure 4 molecules-26-07170-f004:**
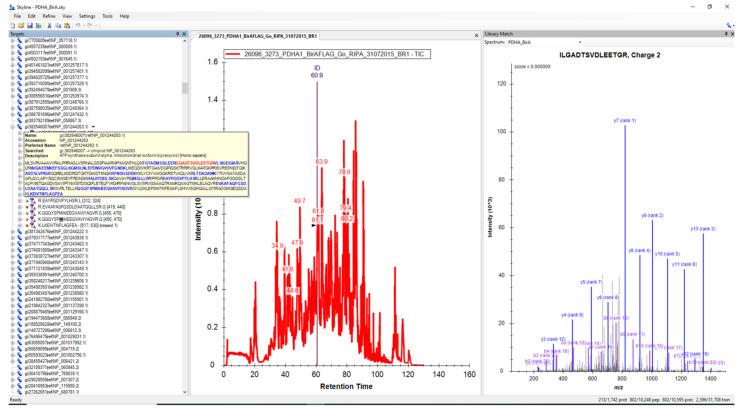
Results of analysis of data from paper Go et al. [45] using Skyline program. All peptides are grouped into lists from interacting or proximal proteins on the left side, the total ion chromatogram (TIC) in the center and MS/MS spectra are shown in the right part of this figure. Example of ATP synthase (inset on left side) as a protein, proximal to PDHA1.

## Data Availability

The MS proteomics data have been deposited into the ProteomeXchange Consortium via the PRIDE partner repository, with the data set identifier PXD015756.

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
