# Peer review of "Application of Skyline for Analysis of Protein–Protein Interactions In Vivo"

_molecules, 2021, doi:10.3390/molecules26237170_

Round 1

Reviewer 1 Report

This is the reviewed manuscript “Application of Skyline for analysis of protein-protein interactions in vivo” where Arman Kulyyassov describes the application of the Skyline software for the analysis of protein-protein interactions by quantifying the biotinylated form of the biotin acceptor peptide tag.

The author had answered and rewritten all the issues previously raised, and therefore, this reviewer thinks it is suitable for publication.

Just few and minor corrections should be considered before publication:

Figure 1 is slight confusing. Text positions, such as the text above the arrow linking figure A to C (which mentions about 1. Ni-agarose, 2. Propionylation etc) should be better resized and positioned.

Figure 1B. the confocal microscopy figure seems to be taken from someone’s work. It shows arrows with no proper meaning related to this work. Author should therefore provide a self-made cartoon representing the image of a confocal microscopy image or generate himself any image.

Figure 1D. What does the “KxBiotinilated BAP” mean?

The normalization and the calculation of the normalization coefficient mentioned in lines 114-116 should be better explained and depicted.

Author Response

The author thanks very much the reviewer for his time to review the manuscript and his advises to improve it. I carefully followed these advises and I believe that the manuscript is now fairly improved.

Figure 1 is slight confusing. Text positions, such as the text above the arrow linking figure A to C (which mentions about 1. Ni-agarose, 2. Propionylation etc) should be better resized and positioned.

Reply: This correction has been done.

Figure 1B. the confocal microscopy figure seems to be taken from someone’s work. It shows arrows with no proper meaning related to this work. Author should therefore provide a self-made cartoon representing the image of a confocal microscopy image or generate himself any image.

Reply: This correction has been done.

Figure 1D. What does the “KxBiotinilated BAP” mean?

Reply: This was corrected in Figure 1 by adding formula and explanation in the text. 

The normalization and the calculation of the normalization coefficient mentioned in lines 114-116 should be better explained and depicted.

Reply: In the control (BAP-Y+BirA-Z) and in the experiment (BAP-Y+BirA-X), as well as in the replicates, the expression levels of recombinant BAP-Y proteins may differ. Variations in the total amount of BAP-Y can also appear during sample preparation. Thus direct comparison of the biotinylation levels AbBAP between samples is not correct. Therefore, the total amount of BAP-Y including its biotinylated and propionylated forms in all samples was normalized and the normalization coefficients were determined. 

Reviewer 2 Report

The author has answered all my questions in the revised manuscript, and I have not found anything that needs to be revised currently.

Author Response

The author thanks very much the reviewer for his time to review the manuscript and his advises to improve it. I carefully followed these advises and I believe that the manuscript is now fairly improved.

This manuscript is a resubmission of an earlier submission. The following is a list of the peer review reports and author responses from that submission.

Round 1

Reviewer 1 Report

Quantitative analysis of protein-protein interactions in living cells is important to elucidate the cell signaling in many biological processes. In this manuscript, the author performed Proximity Utilizing Biotinylation (PUB) based method to validate the interaction of transcription factors Sox2 and Oct4. After normalization and recalculation of data processed by Skyline, high level of biotinylation was detected which demonstrated the feasibility of the method. This research has great value to both the proteomics and the biology community, which would inspire more protein-protein interaction research with biological and clinical application prospect. Besides these principal concerns I would like to point out several other minor points that the authors should consider: 1. Is Skyline a necessary search software to use in this research? Is there any other softwares can also be used for the PPI quantitative analysis here? 2. The author had better to introduce how to calculate this ionization coefficient k=11.9 in the manuscript. 3. The manufacturer shall be indicated for the experimental materials used. 4. The Materials and Methods part missed procedures for all the experiments, for example the cell culture condition, cell lysis method, propionylation method, the tryptic procedures, and the LC-MS/MS parameters, etc.

Reviewer 2 Report

Protein-protein interaction (PPI) analysis is an important and growing field. Allied to the development of different methodologies and equipment such as mass spectrometry have allowed one to better characterize this phenomenon and therefore, generated several new data in different areas of the biology.

In this manuscript, Arman Kulyyassov describes the application of the Skyline software for the analysis of protein-protein interactions by quantifying the biotinylated form of the biotin acceptor peptide tag. For this, the author incorporates in both Skyline and Mascot database, a biotinilated precursor form of the biotin acceptor peptide (BAP). The author analyze the interaction of BAP with humanized biotin ligase BirA peptide by identifying and quantifying HEK293T cells coexpressing the recombinant BAP-Sox2 and BirA-Oct4 interacting proteins or BAP-GFP as negative control.

Although the author show the Sox2-Oct4 interaction by the higher level of biotinylation compared to the GFP-BAP control, I think this work is rather too preliminary for the publication in this journal. The author use just one example using two well-known interacting proteins (Sox2 and Oct4) being expressed in HEK293T cells to state that the Skyline software is suitable for the analysis of protein-protein interactions in vivo. Although this statement may be true, the author need to better develop and present more data to confirm this statement.

No significant results other than the chromatogram shown in Fig 3A are presented. In addition, peak area values corresponding to the biotinylated forms of the peptide are not shown. Moreover, the graphic bars shown on the right side of the chromatograms show error bars, but author do not mention the number of experiments performed, neither the number of spectra were analyzed to draw this graph.

Description on the materials and methods section is just a description on how to generate the MGF file and input the database in Mascot.

As a minor concern, in the first line of the introduction section (line 24), author mention about recent works but cites article from 2013. Author should mention a more recent work.

Finally, author may have used data generated recently by Go and coworkers: A proximity-dependent biotinylation map of a human cell. Nature 2021 (https://www.nature.com/articles/s41586-021-03592-2) and test the possibility of the analysis using Skyline.

In conclusion, because of all above mentioned points, this reviewer thinks that this manuscript does not reach the level for publication in this journal.